# Scaling Laws of Distributed Random Forests

**Katharina Flügel**                                                    *katharina.fluegel@kit.edu*
*Karlsruhe Institute of Technology (KIT), Scientific Computing Center (SCC)*
*Helmholtz AI*

**Charlotte Debus**                                                     *charlotte.debus@kit.edu*
*Karlsruhe Institute of Technology (KIT), Scientific Computing Center (SCC)*

**Markus Götz**                                                          *markus.goetz@kit.edu*
*Karlsruhe Institute of Technology (KIT), Scientific Computing Center (SCC)*
*Helmholtz AI*

**Achim Streit**                                                         *achim.streit@kit.edu*
*Karlsruhe Institute of Technology (KIT), Scientific Computing Center (SCC)*

**Marie Weiel**                                                          *marie.weiel@kit.edu*
*Karlsruhe Institute of Technology (KIT), Scientific Computing Center (SCC)*
*Helmholtz AI*

**Reviewed on OpenReview:** *https://openreview.net/forum?id=ICHxTlgnSy*

## Abstract

Random forests are a widely used machine learning technique valued for their robust predictive performance and conceptual simplicity. They are applied in many critical applications and often combined with federated learning to collaboratively build machine learning models across multiple distributed sites. The independent decision trees make random forests inherently parallelizable and well-suited for distributed and federated settings. Despite this perfect fit, there is a lack of comprehensive scalability studies, and many existing methods show limited parallel efficiency or are tested only at smaller scales. To address this gap, we present a comprehensive analysis of the scaling capabilities of distributed random forests on up to 64 compute nodes. Using a tree-parallel approach, we demonstrate a strong scaling speedup of up to 31.98 and a weak scaling efficiency of over 0.96 without affecting predictive performance of the global model. Comparing the performance trade-offs of distributed and local inference strategies enables us to simulate various real-life scenarios in terms of distributed computing resources, data availability, and privacy considerations. We further explore how increasing model and data size improves prediction accuracy, scaling up to 51 200 trees and 7.5 million training samples. We find that while distributing the data across nodes leads to super-scalar speedup, it negates the predictive benefit of increased data. Finally, we study the impact of distributed and non-IID data and find that while global imbalance reduces performance, local distribution differences can help mitigate this effect.

## 1 Introduction

Interpretability and transparency are crucial for machine-learning applications in fields like healthcare and critical infrastructure, where decisions must be both data-driven and explainable. In such high-stakes domains, black-box models like deep neural networks often face skepticism due to their lack of transparency. As a result, practitioners often prefer to rely on more interpretable alternatives like decision trees. Random forests (Breiman, 2001) are ensembles of independent decision trees. By aggregating the predictions of individual trees, they improve generalization, enhance robustness to noise, and mitigate overfitting. Combining

conceptual simplicity with competitive performance, especially on high-dimensional tabular data, they have been employed in various fields like healthcare (Khalilia et al., 2011; Nguyen et al., 2013; Abdoh et al., 2018; Kaur et al., 2019; Alam et al., 2019; Wang et al., 2020), environmental sciences (Zhang & Yuan, 2015; Benali et al., 2019; Tyralis et al., 2019; Asgari et al., 2022; He et al., 2022), remote sensing (Belgiu & Drăguţ, 2016), material sciences (Li et al., 2022), intrusion detection (Masarat et al., 2016; Resende & Drummond, 2018; Liu et al., 2021), and finance (Madaan et al., 2021; Uddin et al., 2022).

In light of the rising awareness around data security and privacy, federated learning (McMahan et al., 2017; Yang et al., 2019) has emerged as a promising approach for applications involving personal data distributed across multiple sites. In federated learning, multiple independent clients collaboratively train a machine learning model with limited data sharing between participants. Critical applications, such as hospitals and financial institutions, often combine federated learning with random forests (Liu et al., 2020; Hauschild et al., 2022). As ensembles of independent trees, random forests are inherently well-suited for parallelization and distribution. This enables the training of larger forests on more extensive datasets. However, many existing implementations suffer from suboptimal parallel efficiency and often lack a thorough investigation of scaling capabilities. Furthermore, the data used in federated learning is typically assumed to be independent and identically distributed (IID) over all participating clients. In practice, this assumption rarely holds, as the local data distributions can vary significantly across the clients. Understanding how non-IID data affects the performance of distributed machine learning models is crucial for enhancing their practical application in federated scenarios.

This paper aims to address these gaps with a comprehensive study of the scalability and data heterogeneity in distributed random forests. To this end, we utilize a hybrid parallel implementation of distributed random forests, combining the message-passing interface `MPI` with shared-memory parallelism to enable efficient communication and scalability in distributed-memory environments. In a comprehensive scalability study, we thoroughly investigate the scaling capabilities of distributed random forests with respect to the number of samples, trees, and machines, as well as their impact on prediction accuracy. We demonstrate the strong and weak scalability of this approach, utilizing model parallelism by distributing subforests and optionally data parallelism by distributing the training data. We explore when distributing trees and samples is beneficial and how it impacts the final model's prediction accuracy. By comparing different inference flavors in terms of efficiency and memory consumption, we simulate various real-life scenarios in terms of distributed compute power, data availability, and privacy restrictions. Finally, we study the impact of data imbalance on federated learning scenarios by treating each machine as an individual client with its local data. To explore the impact of non-IID data on the predictive performance of federated random forests, we evaluate the effect of class imbalance in both global and local datasets. This provides deeper insights into how random forests behave in federated learning settings and their robustness to heterogeneous data distributions.

Our key contributions are:

- An experimental study of the scaling laws of random forests with increasing model and data size.
- An HPC-adapted, hybrid parallel implementation of distributed random forests.
- A parallel scalability study on up to 64 compute nodes, achieving up to 31.98 strong scaling speedup and over 0.96 weak scaling efficiency without affecting predictive performance of the global model.
- An evaluation of the effects of data distribution and non-IID class distributions on distributed random forests.

## 2 Related Work

**Parallel Random Forests** Multiple shared-memory parallel implementations of random forests in `R` have been suggested (Genuer et al., 2017; Wright & Ziegler, 2017; Azizah et al., 2019), which parallelize over the trees on up to 15 cores, but lack a thorough evaluation of parallel scalability. Random forests have also been parallelized using accelerators such as GPUs and FPGAs. For instance, Van Essen et al. (2012) parallelize the inference over the sample domain, while Senagi & Jouandeau (2022) parallelize the training over the trees, achieving average speedups of up to 3.57 with dynamic GPU parallelism.

**Distributed Random Forests with MapReduce**  Many approaches apply `MapReduce` (Dean & Ghemawat, 2008) to distribute random forests across multiple worker nodes. They differ in the exact frameworks used, the computational partitioning, and the evaluation scale, often utilizing only a small number of parallel nodes. Panda et al. (2009) parallelize best-split selection along tree nodes using `MapReduce` and report strong scaling results for training a single tree on up to 200 workers. Basilico et al. (2011) propose a `MapReduce`-based parallelization over the trees of the forest, assigning each worker node its own data subset. They evaluate their approach on 65 parallel worker nodes with four cores each, reaching speedups of up to ten, but do not report detailed parallel scaling results. Han et al. (2013) implement another parallelization along the tree dimension. They evaluate their approach on up to ten workers in a weak-scaling experiment by scaling up the number of trees with the workers, but do not report timing or speedup results. Masarat et al. (2016) and Liu et al. (2021) apply a similar approach to intrusion detection. Zhang & Yuan (2015) use `MapReduce` to parallelize along the sample dimension, first computing local splitting criteria before aggregating them. They evaluate their approach on up to eight cores across four dual-core CPUs but achieve no significant parallel speedup. This is also confirmed by Asgari et al. (2022), who report speedups of at most 1.1 across four compute nodes using the same approach. Wakayama et al. (2015) train random forests with large datasets on `MapReduce` clusters, where each node holds a local training dataset. To address overfitting due to limited data access per node, they construct "shared forests" on the manager node. These shared forests are adapted to the local training data on each cluster node using transfer learning, and then returned to the manager for further processing. Chen et al. (2017) proposed a random forest algorithm that combines data and task parallelism using a task-directed acyclic graph. The data is partitioned vertically along the feature axis. During training, trees are constructed in parallel, with multiple feature variables in each tree calculated concurrently for node splitting. They evaluate their approach on up to 100 parallel workers with two cores each and report speedups of up to 87.3 at 100 worker nodes.

**Other Approaches to Distribute Random Forests**  Most closely related to our work are approaches using general distributed computing in high-performance computing (HPC) environments. Mitchell et al. (2011) introduce a manager-worker-based Parallel Random Forest Classifier for `R`, distributing bootstrap samples and trees among available parallel processes. Communication is facilitated through a custom reduction function, akin to `MPI`. They report parallel efficiency over 0.5 on up to 128 cores for strong scaling and 512 cores for weak scaling with increasing forest size. Wang et al. (2018) introduce *DistForest*, a parallel random forest for supercomputers using `MPI`. Their manager-worker-based approach distributes dataset features across the parallel workers, utilizing multiple manager nodes to facilitate parallel tree training. *DistForest* is evaluated on up to 128 nodes; however, the parallel efficiency remains below 0.5. Cliff et al. (2019) implement iterative random forests, which add an iterative boosting process to standard random forests, for HPC applications using `C++` and `MPI`. Their parallelization approach aligns with the shared-global-model variant in our paper, where subforests are distributed across compute nodes and then aggregated into a shared model. Similar to Mitchell et al. (2011), they focus on the application to biological datasets where the features often outnumber the samples. They evaluate an iterative random forest for gene expression on up to 1600 threads across ten nodes. However, their parallel evaluation is limited to a single strong-scaling setup with 1000 trees, and parallel efficiency drops below 0.5 after five nodes. Vázquez-Novoa et al. (2023) use the task-based programming model COMPs to apply task parallelism along both the tree and data dimensions. They evaluate strong and weak scaling with an increasing number of trees on up to 16 compute nodes, each with 48 cores, reporting strong scaling speedups of up to six; weak scaling efficiency at 16 nodes is approximately 0.65.

**Federated Learning**  Distributed random forests are closely related to federated learning (McMahan et al., 2017; Yang et al., 2019), where multiple participants collaboratively train a model without sharing their data, often motivated by concerns over data privacy and security. Federated learning can be categorized into three types: horizontal federated learning, where all participants share the same features but different samples; vertical federated learning, where the participants hold different features of the same samples; and federated transfer learning, where there is no overlap between either features or samples. With the independence of individual trees, random forests lend themselves to federated learning, and multiple approaches have been introduced for both horizontal and vertical federation. Most random-forest-based approaches for horizontally partitioned data first train independent subforests on each participant's local data and then aggregate them

into a global ensemble. Different aggregation approaches have been introduced, such as weighting the trees by their performance (Tsou et al., 2018; Gencturk et al., 2022) and selecting only a subset of all trees (Markovic et al., 2022; Hauschild et al., 2022). Most approaches for tree-based vertical federated learning focus on growing decision trees collectively, node by node, to find the best split across all features. These approaches can be adapted to random forests by applying the same approach to multiple trees. Liu et al. (2020) introduce *Federated Forest* for vertical federated learning. In each node, clients provide potential splits and corresponding scores, while a central server determines the best split based on these scores. Wu et al. (2020) have clients collectively determine the best split using multi-party computation and homomorphic encryption. *FedTree* (Li et al., 2023) adopts a similar approach to *Federated Forest* (Liu et al., 2020) for training gradient-boosted decision trees in both vertical and horizontal federated learning settings. Clients send local histograms to a central server, which aggregates these histograms using summation (for horizontal federation) or concatenation (for vertical federation) and decides on the split in the current node.

**Non-IID Data**   Many distributed random forest approaches assume the data to be independent and identically distributed (IID) across all partitions. However, in practical federated applications, the local data subsets are often non-IID, which can pose an additional challenge (McMahan et al., 2017; Lu et al., 2024). There are multiple ways in which the local subsets can differ, ranging from differences in sheer size to differences in feature and class distributions, both individually and jointly. Multiple categorizations have been suggested (Criado et al., 2022; Ma et al., 2022; Milasheuski et al., 2024). In this paper, we focus on a common type of non-IID data: varying class (im)balances, where the share of different classes varies between the subsets. Multiple studies examine the effects of both generally imbalanced and non-IID data on the predictive quality of random forests. However, most approaches focus on the special case of binary classification and consider either an imbalance in the global dataset or between the local subsets, but not their interaction. For example, Hauschild et al. (2022) investigate how the number of data splits, imbalances in local data sizes, and global class imbalances affect the performance of random forests for binary classification in horizontal federated learning. Antonio Eng Lim & Hee Park (2024) build a federated random forest by growing each tree collaboratively across clients. They evaluate the impact of varying class imbalance between the local subsets for multi-class classification while keeping the global class balance fixed. Shen et al. (2022) study the effects of class imbalance for image classification with neural networks, suggesting an optimization-based meta-algorithm to improve performance in the minority classes through iterative communication rounds. The evaluate their approach combining both global imbalance using up to three minority classes with a local imbalance following the approach by Hsu et al. (2019).

**Scaling Laws**   Neural scaling laws (Kaplan et al., 2020) study how the performance of machine learning models improves as a function of compute, data, and model size. Kaplan et al. (2020) demonstrate empirically that the performance of language models scales as a power law with the number of parameters, dataset size, and amount of compute. Multiple follow-up studies (Sharma & Kaplan, 2022; Michaud et al., 2023; Bahri et al., 2024) examining the theory behind it. While initially focused on transformer-based language models, they have since been extended to other architectures and applications like image generation and classification (Henighan et al., 2020; Alabdulmohsin et al., 2022), reinforcement learning (Gao et al., 2023), and time-series forecasting (Shi et al., 2024). For random forests, Biau (2012) gives a theoretical analysis of how their performance scales with the number of features and shows that the convergence is independent of the number of noise features, and multiple studies (Oshiro et al., 2012; Probst & Boulesteix, 2018) analyze how the predictive performance scales with the number of trees. We extend this analysis with an empirical evaluation of the scaling with both the number of trees and samples.

## 3   Background: Random Forests

A random forest (Breiman, 2001) $F = \{T_1, \ldots, T_t\}$ is an ensemble of $t$ decision trees $T_i$. Random forests can be applied to both classification and regression tasks, yielding robust and accurate predictions for many applications. The fundamental building blocks of random forests are the decision trees $T_i$, which recursively partition the feature space. Multiple algorithms exist for constructing decision trees, which can be combined with random forests. These include CART (Breiman, 1984), ID3 (Quinlan, 1986), AID20 (Morgan et al., 1963), and THAID (Morgan & Messenger, 1973). To train a random forest on a dataset $D$ consisting of

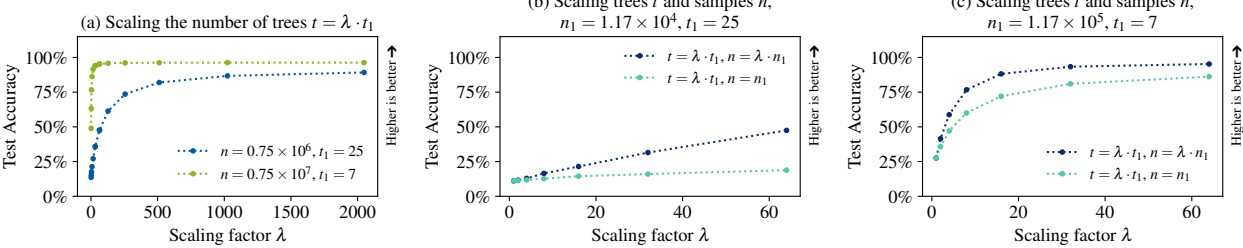

Figure 1: What is the impact of the model size (number of trees $t = \lambda \cdot t_1$) and training data (samples $n$) on the predictive performance of random forests? (a) When increasing the model size $t$ while keeping the data size $n$ constant, we observe a consistent improvement from adding more trees that slowly saturates for both baselines $t_1 = 25$ with $n = 0.75 \times 10^6$ and $t_1 = 7$ with $n = 0.75 \times 10^7$. (b, c) The accuracy benefits further from simultaneously scaling the dataset to $n = \lambda \cdot n_1$ samples compared to increasing only the model size and keeping the data static at $n = n_1$. More details on the experimental setup are given in Section 5.

$n$ samples with $m$ features, each tree is trained individually by growing from its root node. At each node, only a random subset of $u$ features is considered for splitting ("feature bagging"), often set to $u = \sqrt{m}$. This helps reduce correlation among trees and improve generalization. Among the selected features, the best split is chosen to divide the samples at a node into subsets while maximizing the purity of the subsets. Multiple optimization metrics exist to measure this impurity, such as the Gini index, used in this paper, and information gain. This splitting is repeated in the child nodes until a stopping criterion is reached, for example, a certain purity in the leaf nodes. The leaf node is assigned a prediction based on voting or averaging over the remaining training samples within its region. Bootstrapping is used to create multiple training datasets by randomly sampling a bootstrap $D_i$ with replacement from the original dataset $D$ for each tree $T_i$, typically $|D_i| = n$. This process is known as bagging (bootstrap aggregating) and introduces diversity among the trees in the forest. Overall, the complexity of growing a single tree up to depth $\mathcal{O}(\log n)$, considering $n$ samples and $u$ features at each split, can be summarized as $\mathcal{O}(u \cdot n \log n)$. The $t$ trees are trained independently, resulting in $\mathcal{O}(t \cdot u \cdot n \log n)$ to train the entire forest. During inference, a sample is processed by all decision trees. Each tree makes a prediction by passing the sample from the root to the corresponding leaf node. The random forest aggregates the individual predictions into the ensemble result.

## 4 Distributed Random Forests

The predictive performance of random forests improves with both the size of the random forest and the amount of data it is trained on, as demonstrated in Figure 1. However, with growing datasets and models, the computational and memory demands can surpass the capacity of a single machine. This highlights the need for distributing random forests across multiple machines. Yet, there is a distinct lack of comprehensive studies on the scalability of distributed random forests. Since the individual decision trees of the random forest are independent, they are an ideal target for parallelization. We therefore partition the $t$ trees and distribute them equally among the processing units. During training, each node holds a local subforest $F_{\text{local}}$ of $t/p$ trees and trains them using either shared global data or node-local data $D$. Training the subforests does not require any communication between compute nodes. Figure 2(a) illustrates this parallel training process. As most modern compute nodes follow a multi-core architecture, it is often beneficial to further parallelize over the $t/p$ local trees within each node's subforests. We thus employ a hybrid parallelization scheme, using distributed computing between the $p$ compute nodes and shared-memory parallelization within each node.

For global inference, a sample needs to be processed by all trees; however, during training, the global ensemble is distributed across compute nodes. We consider two different approaches for inference on a distributed random forest. First, one can aggregate a **global model** by collecting the local subforests $F_{\text{local}}$ and constructing the global forest $F_{\text{global}} = \bigcup_{p}^{i=1} F_{\text{local}}(i)$. Afterwards, the global model can be used independently of other nodes to conduct inference for arbitrary, potentially local samples (see Figure 2(b)). This can be advantageous when local test data cannot be shared among nodes, such as in federated applications. However,

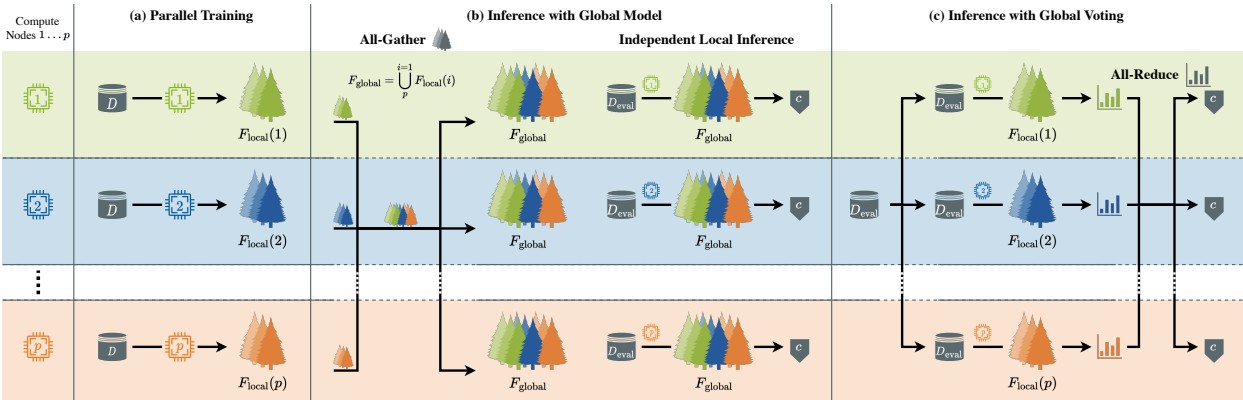

Figure 2: Example of a parallel random forest using compute nodes $1, 2, \ldots p$ each node $i$ training three trees in their local subforests $F_{\text{local}}(i)$, for a global forest $F_{\text{global}}$ of nine trees. (a) Parallel Training: Node-local subforests are trained independently of other nodes, either on shared global data or local slices $D$. (b) Inference with Global Model: Aggregate local subforests to global model after training, allowing local inference independent of other nodes. (c) Inference with Global Voting: Share evaluation data $D_{\text{eval}}$, perform distributed inference on local subforests, and aggregate local results to global prediction.

aggregating a global model requires a significant one-time communication overhead for gathering all local subforests and does not benefit further from distribution in the inference phase. Alternatively, the compute nodes can continue collaboration during the inference phase via **global voting**, as illustrated in Figure 2(c). Each compute node continues to hold only its local subforest $F_{\text{local}}$. Test samples are processed in parallel on all local subforests. To aggregate the results to the global prediction, each subforest computes a histogram of its votes. These histograms are summed using collective reduction, after which the global prediction can be determined from its maximum. With this approach, the predictions are identical to those obtained by an aggregated global model. At the same time, it continues to benefit from distributed computing in the inference phase, which both speeds up the inference and reduces the memory requirement on each node. This, however, requires the test data to be shared among all nodes, which may violate data privacy requirements in some applications. The pseudocodes for distributed training and inference are given in Appendix A.1.

## 5 Empirical Results

### 5.1 Datasets

We evaluate the parallel scalability of distributed random forests in terms of both strong and weak scaling and their application to federated learning. Details on the implementation and computational environment are given in Appendix A.2. We use synthetic data generated with `scikit-learn`'s `make_classification` as this allows us to scale both the number of samples $n$ and features $m$ freely and adjust the class balance. We keep the fraction of informative and redundant features at $10\%$ each, with $0\%$ repeated features, and all remaining features filled with noise. The number of classes is set to ten, with one cluster per class. Except for the experiments with imbalanced data in Section 5.5, we use balanced classes. For most experiments, we use the largest dataset we could fit together with the local model onto a standard compute node, resulting in a total of $1 \times 10^{10}$ values. We use two datasets with different feature-to-sample ratios, resulting in two distinct difficulties. The **1M** dataset consists of $n_{\text{total}} = 1 \times 10^6$ samples and $m = 1 \times 10^4$ features, while the **10M** dataset consists of $n_{\text{total}} = 1 \times 10^7$ samples and $m = 1 \times 10^3$ features. The **1M** dataset is more challenging to solve, as it has an order of magnitude more features to learn from fewer samples. When comparing the two inference variants, we need to fit both the data and the global model into the memory of individual compute nodes. We thus use two additional, smaller datasets with only $1 \times 10^8$ overall values, as the $1 \times 10^{10}$ value datasets exceed our memory capacity. The **100K** dataset has $n_{\text{total}} = 1 \times 10^5$ samples and $m = 1 \times 10^3$ features, while the **1M-b** dataset has $n_{\text{total}} = 1 \times 10^6$ samples and $m = 1 \times 10^2$ features. Additionally, we

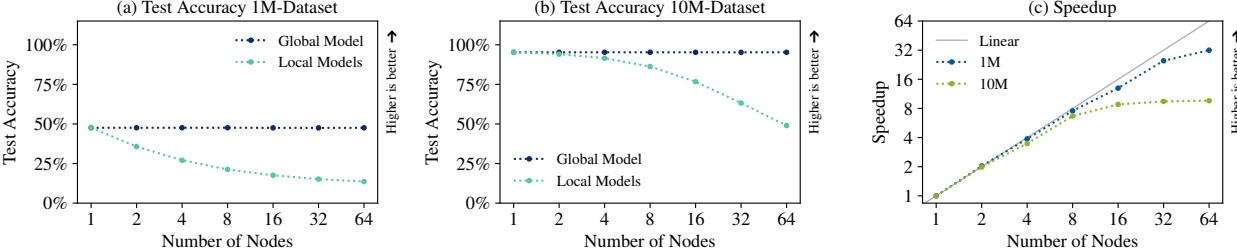

Figure 3: Strong scaling experiments: two fixed size problems training $t = 1600$ trees on the **1M** dataset and $t = 448$ trees on the **10M** dataset while increasing the number of compute nodes $p$. Each node trains a subforest of size $t/p$. The global accuracy is not affected by parallelization and remains constant. The speedup initially increases linearly but saturates as the work assigned to each node becomes too small for effective parallelization.

extend the strong and weak scaling experiments to the HIGGS dataset (Baldi et al., 2014; Whiteson, 2014), a binary classification task to distinguish between signal and background events in particle collision data. The dataset is nearly balanced, containing about $53\%$ positive (signal) samples. With $n_{\text{total}} = 1.1 \times 10^7$ samples and $m = 28$ features, it is similar in size to the 10M synthetic dataset. For all datasets, $75\%$ of the samples $n = 0.75 \cdot n_{\text{total}}$ are used as training set, while the remaining $25\%$ are used as test set. During bootstrapping, each tree draws a random set of $n$ samples with replacement.

## 5.2 Scalability of Distributed Training

To evaluate the parallel scalability of distributed random forests, we use two common types of scaling experiments: strong and weak scaling. In strong scaling, the goal is to solve a fixed problem faster by increasing the computing resources. In our case, we train a random forest using a fixed number of trees $t$ on the same dataset while increasing the number of compute nodes. We run two series of experiments: training $t = 1600$ trees on the **1M** dataset and $t = 448$ trees on the **10M** dataset on $p \in \{1, 2, 4, 8, 16, 32, 64\}$ compute nodes. Each node trains a local forest of $t/p$ trees. The number of trees was chosen as the maximum multiple of 64 we could train within $100\,\text{min}$ on a single node. For the serial baseline with $p = 1$, we train a `scikit-learn RandomForestClassifier` on all $t$ trees without `MPI` but with the shared-memory parallelization. Note that since the full-sized global forest does not fit onto a standard compute node, the serial runs were conducted on high-memory nodes with twice the main memory but otherwise identical hardware. Figure 3 illustrates the results in terms of test accuracy and speedup; detailed results of training time and speedup are given in Table 1. All accuracies are given as the mean over three independent runs with separate random seeds passed to the model. As expected, the accuracy of the global model remains unchanged when increasing $p$, as this does not affect the global number of trees $t$. This confirms that our parallelization does not impact the random forest's predictive performance. Comparing the two datasets, we observe that the model performs significantly better on the **10M** dataset than on the **1M** dataset, even with considerably fewer trees in the forest. This is expected, as the ratio of samples to features is 100 times higher for **10M** than for **1M**, making it harder to learn. In contrast to the global model, the mean accuracy of the local models decreases with $p$ as their size $t/p$ decreases. Finally, we evaluate the parallel scalability using the speedup $S(p) = T_{\text{seq}}/T(p)$, where $T_{\text{seq}}$ is the training time of the sequential baseline and $T(p)$ is the parallel training time with $p$ compute nodes. Ideally, the speedup would scale linearly with the number of compute nodes $p$. Our results show linear speedup initially, but this saturates as the number of nodes increases. This effect is more pronounced for the **10M** dataset compared to the **1M** dataset. This is expected as the work assigned to each node reduces with increasing $p$. At $p = 64$, the subforests on each node contain only 25 (**1M**) and 7 (**10M**) trees, limiting the gain from further parallelization.

In weak scaling, we aim to solve a larger problem by increasing the computing resources while the problem size per node remains constant. In our case, we scale up the number of trees $t$ as we increase the number of compute nodes $p$. Specifically, we train a global forest with $t = t_1 \cdot p$ trees, where $t_1$ is the baseline number of trees, and each node trains a local subforest with $t_1$ trees. As for strong scaling, we train a

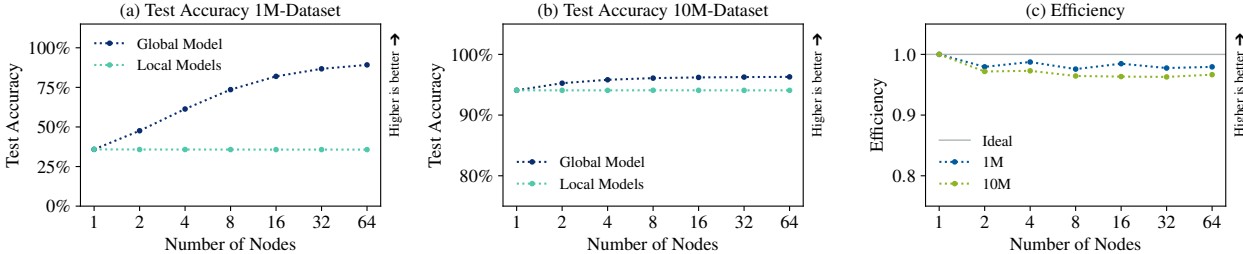

Figure 4: Weak scaling experiments: increasing the global number of trees $t = t_1 \cdot p$ in the random forest while increasing the number of compute nodes $p$ for two baseline problems, training $t_1 = 800$ trees on the **1M** dataset and $t_1 = 224$ trees on the **10M** dataset. Each node trains a subforest of size $t_1$. The global accuracy increases with $p$ as the number of trees in the global forest increases. The parallel efficiency remains over 0.96 for both datasets and all tested scales $p$.

Table 1: The global training time $T(p)$, strong scaling speedup $S(p)$, and weak scaling efficiency $E(p)$ for $p = 1, \ldots, 64$ nodes with the corresponding number of cores, trees $t$, and trees per node $t/p$.

| | Nodes $p$ | Cores | **Strong Scaling** | | | | **Weak Scaling** | | | |
|---|---|---|---|---|---|---|---|---|---|---|
| | | | Trees $t$ | $t/p$ | $T(p)$ ↓ | $S(p)$ ↑ | Trees $t$ | $t/p$ | $T(p)$ ↓ | $E(p)$ ↑ |
| **1M** | 1 | 76 | 1600 | 1600 | 5923.104 s | 1.000 | 800 | 800 | 2870.629 s | 1.000 |
| | 2 | 152 | 1600 | 800 | 2911.042 s | 2.035 | 1600 | 800 | 2930.780 s | 0.979 |
| | 4 | 304 | 1600 | 400 | 1528.165 s | 3.876 | 3200 | 800 | 2907.165 s | 0.987 |
| | 8 | 608 | 1600 | 200 | 784.943 s | 7.546 | 6400 | 800 | 2941.895 s | 0.976 |
| | 16 | 1216 | 1600 | 100 | 456.719 s | 12.969 | 12800 | 800 | 2915.327 s | 0.985 |
| | 32 | 2432 | 1600 | 50 | 238.113 s | 24.875 | 25600 | 800 | 2936.716 s | 0.977 |
| | 64 | 4864 | 1600 | 25 | 185.237 s | 31.976 | 51200 | 800 | 2930.885 s | 0.979 |
| **10M** | 1 | 76 | 448 | 448 | 5839.393 s | 1.000 | 224 | 224 | 2881.014 s | 1.000 |
| | 2 | 152 | 448 | 224 | 2938.522 s | 1.987 | 448 | 224 | 2964.515 s | 0.972 |
| | 4 | 304 | 448 | 112 | 1681.794 s | 3.472 | 896 | 224 | 2961.187 s | 0.973 |
| | 8 | 608 | 448 | 56 | 874.932 s | 6.674 | 1792 | 224 | 2987.388 s | 0.964 |
| | 16 | 1216 | 448 | 28 | 662.736 s | 8.811 | 3584 | 224 | 2990.356 s | 0.963 |
| | 32 | 2432 | 448 | 14 | 617.535 s | 9.456 | 7168 | 224 | 2992.089 s | 0.963 |
| | 64 | 4864 | 448 | 7 | 606.976 s | 9.620 | 14336 | 224 | 2980.575 s | 0.967 |

forest with $p \in \{1, 2, 4, 8, 16, 32, 64\}$ nodes on the **1M** and **10M** datasets. We use $t_1 = 800$ for **1M** and $t_1 = 224$ for **10M**. That is, the global forests trained by strong scaling correspond to those trained at $p = 2$ in the weak scaling experiments. Figure 4 and Table 1 give our results. In contrast to strong scaling, the accuracy of the local forests remains constant while the global forest improves as we increase the number of compute nodes $p$ and, in turn, the global model size $t = t_1 \cdot p$. The improvement in global accuracy is significantly higher for the **1M** compared to the **10M** dataset, as there is much more room for improvement from the $t_1$ baseline. Since the problem size increases with $p$, we measure the parallel performance in terms of weak scaling efficiency $E(p) = T_{\text{seq}}/T(p)$ instead of speedup. Efficiency typically ranges from zero to one, with higher values indicating better performance and one being the expected ideal. Our implementation demonstrates good scalability with efficiencies over 0.96 across all scales. In contrast to strong scaling, we observe no saturation as we increase $p$ and scale successfully up to 64 compute nodes (the maximum tested). Since we scale the problem size in tandem with the available resources, we avoid the issue of having too little work left on each node.

Figure 5 gives the strong and weak scaling performance on the HIGGS dataset. We train a global forest of $t = 640$ trees for strong scaling and scale up from $t_1 = 10$ trees in weak scaling, each with up to 64 nodes. The scaling behavior for both predictive and parallel performance matches our results on synthetic data. The accuracy is not affected by the parallelization and depends only on the model size. In strong scaling,

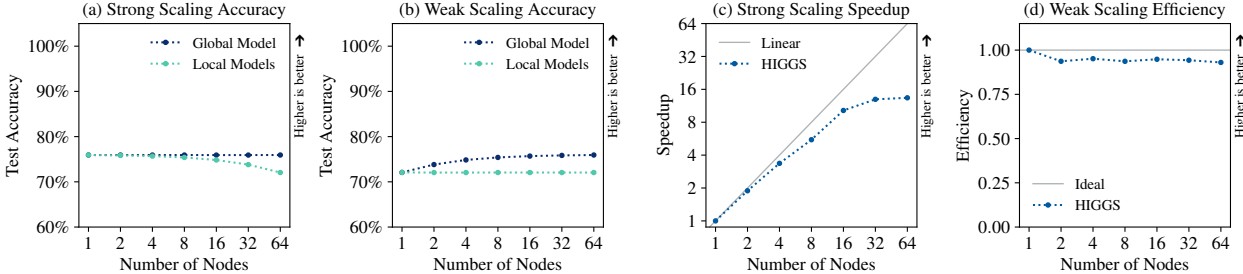

Figure 5: Strong scaling with $t = 640$ global trees and weak scaling with $t_1 = 10$ trees per node on the HIGGS dataset. The results mirror the behavior on synthetic data: The accuracy is not affected by parallelization and depends only on model size. The strong scaling speedup initially increases linearly but saturates as the work per node decreases, while weak scaling efficiency remains high throughout all scales.

the accuracy of the global model remains constant while that of the local models decreases slightly as they become smaller. In weak scaling, the local models have constant size and accuracy while the accuracy of the global model improves with more nodes, and thus trees. As with the synthetic data, the strong scaling speedup increases linearly at first but saturates after about 16 nodes. Similarly, the weak scaling efficiency remains close to one at all tested scales.

## 5.3 Scalability of Distributed Inference

We compare the two approaches to inference introduced in Section 4 in terms of their time and memory consumption. With **global voting**, each node $i$ holds only the local subforest $F_{\text{local}}(i)$ it trained. For inference on the global model, each test sample must be shared across all nodes, which perform independent inference on local subforests. The resulting local predictions are then aggregated via voting to obtain the global result. In contrast, building a shared **global model** would aggregate the local subforests $F_{\text{local}}$ once after training and combine them into a global model $F_{\text{global}} = \bigcup_p^{i=1} F_{\text{local}}(i)$ on each node. For inference, each node can use its own global model independent of the other nodes, meaning test data does not need to be shared among nodes. This approach is also motivated by federated learning, where clients may not want to send real test data to others due to privacy concerns. Sharing the model allows each individual node to act independently of the others during the inference phase. In contrast, global voting would be more suited to a single stakeholder distributing the random forest for computational performance, as in an HPC environment. Note that they do not affect the predictive performance of the global model.

To fit both the data and the global model into the memory of individual compute nodes, we use the smaller datasets **100K** and **1M-b** and $t_1 = 76$ local trees. We use a weak scaling setup, meaning the global model and thus the communication volume increase with $p$. The complexity of all-gathering a message of size $m$ across $p$ nodes is $\mathcal{O}(\alpha \log p + \beta pm)$, where $\alpha$ is the message startup latency and $\beta$ is the communication cost per word (Sanders et al., 2019, Chapter 13). To all-gather the global model, each node sends the $t_1$ trees in its local subforest. The resulting message size $m$ is about $461.97\,\text{MiB}$ for **100K** and $2.11\,\text{GiB}$ for **1M-b**. All-reducing the histograms for global voting has complexity $\mathcal{O}(\alpha \log p + \beta m)$. The size of the histograms $m$ depends on the number of classes and test samples and is about $90\,\text{B}$ per sample for both datasets.

Figure 6 illustrates our results; detailed values are provided in Table 4. First, we consider the overhead of all-gathering the global model after training. Figure 6(a) gives the time to train and aggregate the $t_1 \cdot p$-tree forest on $p$ nodes, normalized with the single-node training time $T_{\text{seq}}$ of a $t_1 \cdot p$-tree forest. As expected, the time to train the local subforest $T_{\text{train}}$ remains constant, independent of the chosen inference variant. Compared to the sequential time, the training time thus decreases with increasing $p$. Aggregating the global model adds an overhead $T_{\text{gather}}$ to the training. This overhead grows with increasing $p$ and thus forest size, while remaining constant with respect to the sequential training time. At $p = 64$ nodes and $t = 4864$ trees, all-gathering the model takes between 3.7 (**1M-b**) and 5.1 (**100K**) times as long as the parallel training itself. Despite this overhead, both variants are significantly faster than sequential training. Figure 6(b) considers the inference time $T_{\text{test}}$ on the global test set, again normalized with the sequential time $T_{\text{seq}}$ on

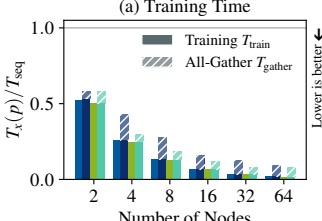 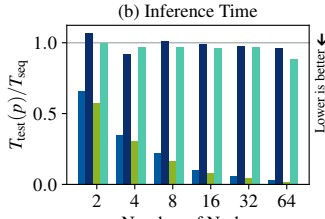 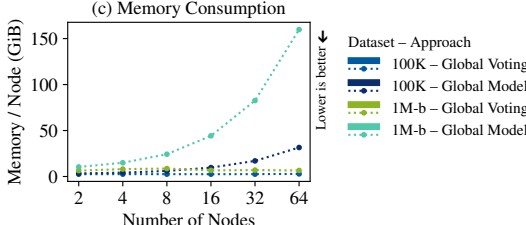

Figure 6: Comparison of two inference variants: aggregating a shared global model on each node after training or performing distributed inference via global voting for two baseline problems, **100K** and **1M-b** with $t_1 = 76$ trees per node in a weak-scaling setup. Building a shared global model comes at both a one-time overhead to aggregate the local models after training and a consistent overhead during the serial inference phase for both inference time and memory consumption. In contrast, global voting requires less time and memory per node but utilizes all $p$ nodes during inference. Training and inference times are normalized with the corresponding single-node time $T_{\text{seq}}$ on the same-size forest with $t_1 \cdot p$ trees.

the corresponding model size. With a shared global model, a node needs to process the entire forest and is thus on par with sequential inference ($T_{\text{test}}(p)/T_{\text{seq}} \approx 1$). This is a significant overhead compared to global voting, which distributes the inference across all $p$ nodes, each processing only their local subforest and thus reducing the inference time with increasing $p$. Finally, Figure 6(c) illustrates the memory consumption per node. With global voting, the memory consumption remains constant, while using a global model increases it with the size of the aggregated global model, growing to more than $150 \, \text{GiB}$. This memory consumption drastically limits the base problem size we could use in this experiment compared to using global voting. Overall, the aggregation of a shared global model comes at both the one-time cost of the all-gathering overhead and the recurring cost of holding and operating on the entire global model. In contrast, global voting reduces the overall wall-clock time and consumes less memory per node. However, it requires using all $p$ nodes and the parallel efficiency can drop below 0.5, especially for small datasets and forests. One thus needs to carefully consider the trade-off between wall-clock time and efficient use of computing resources for the given use case. In some cases, the inference approach can also be prescribed by the application. For example, in federated learning settings, it is often not possible to share local test data with other participants, requiring the entire global model to be held locally. In other cases, the global model may be too big to fit onto a single node, requiring distributed inference.

## 5.4 Training with Distributed Data

In Sections 5.2 and 5.3, we demonstrate the parallel scalability of distributed random forests using shared training data. However, distributed machines may not always have access to the same data. For example, in federated learning scenarios, participants might not wish to share their training data, or data is gathered in different locations with limited communication capacities. We therefore investigate the impact of partitioning the training data across nodes on both the predictive and the computational performance. This approach is related to data parallel training employed for neural networks, where the data is partitioned across machines that collaboratively train a global model (Ben-Nun & Hoefler, 2019). We repeat the strong scaling experiment from Section 5.2, but in addition to the trees, we also partition the training data and distribute it across nodes. Thus, each node trains $t/p$ trees on $n/p$ samples compared to the global $t$ trees and $n$ samples, using the **1M** dataset with $t = 1600$ global trees and the **10M** dataset with $t = 448$ global trees. Figure 7 and Table 5 present the predictive and computational performance for $p \in \{1, 2, 4, 8, 16, 32, 64\}$ compute nodes. For comparison, the corresponding values from the strong scaling experiment (Figure 3) are included with low transparency. For both datasets, the accuracy of the trained forest decreases significantly as the number of nodes increases and the local datasets reduce in size. Compared to strong scaling without data distribution, the accuracy of the local subforests decreases more strongly, and even the global accuracy decreases. With distributed datasets, the bagging of samples on each node is restricted to a much smaller base dataset. This decrease in accuracy is more pronounced for the **1M** dataset. With $n = 0.75 \times 10^6$ samples, only $n/p = 1.17 \times 10^4$ samples per node remain at $p = 64$ nodes—or about 1.17 samples per feature. Due to

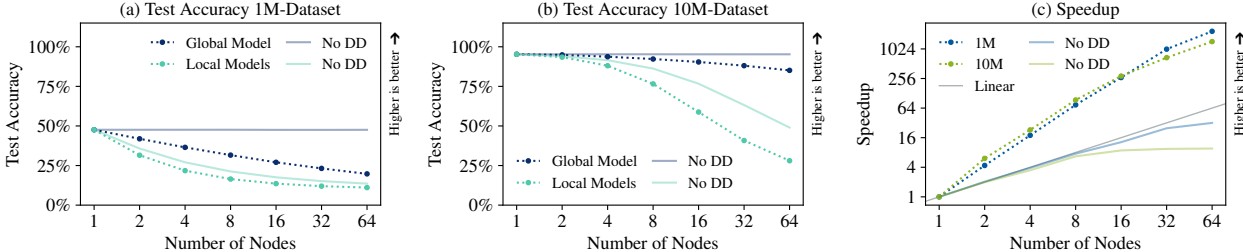

Figure 7: Results for training with distributed data, where the training data is partitioned and distributed over the nodes together with the trees. We solve two fixed-size problems, training $t = 1600$ trees on the **1M** dataset and $t = 448$ trees on the **10M** dataset while increasing the number of compute nodes $p$. Each node trains a subforest of size $t/p$ on $n/p$ samples. The corresponding strong scaling results from Figure 3, i.e., without data distribution (No DD), are superimposed with low transparency. With data partitioning, both local and global accuracy degrade with increasing $p$; this effect is stronger for **1M** with fewer samples overall. However, we observe a super-linear speedup of the overall work decreases with more compute resources.

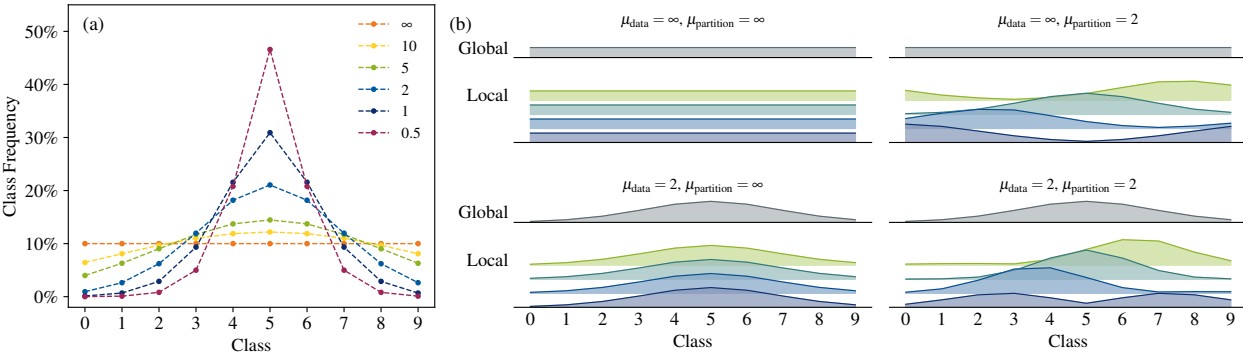

Figure 8: (a) The class frequencies for ten classes with $\mu = \{\infty, 10, 5, 2, 1, 0.5\}$ and a peak at class five. (b) The class frequencies of the global dataset and the different local subsets on $p = 4$ nodes for different combinations of data and partition imbalance. The global data is balanced with $\mu_{\text{data}} = \infty$ in the top row and imbalanced in the bottom row with $\mu_{\text{data}} = 2$. The partition into local subsets is balanced in the left column ($\mu_{\text{partition}} = \infty$) and imbalanced in the right column ($\mu_{\text{partition}} = 2$).

its higher ratio of samples to features, the **10M** dataset retains about 117.18 samples per feature per node at $p = 64$. While the predictive performance decreases, we observe a super-linear speedup in training time. This is expected as the total workload decreases from $\mathcal{O}\left(t \cdot u \cdot n \log n\right)$ to $\mathcal{O}\left(t \cdot u \cdot n/p \log(n/p)\right)$ as each local forest is trained on only $n/p$ instead of $n$ samples.

## 5.5 Breaking the IID Assumption

So far, we have assumed the training data to be independent and identically distributed (IID). However, in many federated settings, this assumption no longer holds, for example, due to different data collection methods or class prevalence. In this section, we examine the behavior of distributed random forests when breaking the IID assumption. Specifically, we vary the class probabilities of both the global dataset and the local subsets independently and examine the impact on the global and local accuracy. We use the probability mass function of the Skellam distribution $p(k; \mu_1, \mu_2) = e^{-(\mu_1 + \mu_2)} \left(\mu_1/\mu_2\right)^{k/2} I_k(2\sqrt{\mu_1 \mu_2})$ (Irwin, 1937; Skellam, 1946) with $\mu = \mu_1 = \mu_2$ to generate different class imbalances, as it is a parametrizable discrete distribution which tends toward the normal distribution $\mathcal{N}(0, 2\mu)$ for large $\mu$ (Irwin, 1937). To obtain the class weights for the $C$ classes $\mathbb{Z}_C = \{0, 1, \ldots, C-1\}$, we sample $p(k; \mu, \mu)$ at $k \in \{c - \lfloor C/2 \rfloor \mid c \in \mathbb{Z}_C\}$. Decreasing $\mu$ thus increases class imbalance. We use the edge case $\mu = \infty$ to indicate balanced classes. Figure 8(a) illustrates the resulting class frequencies for different $\mu$. In this experiment, we consider two different kinds of

class imbalances. First, we consider the imbalance of the global dataset, that is, when generating the global dataset, we use the class frequencies prescribed by $\mu_{\text{data}}$. Second, we vary the imbalance of the data partition into local datasets to create non-IID data, i.e., the class frequencies of the node-local datasets are different across nodes and compared to the global dataset. For this, we use a class distribution with $\mu_{\text{partition}}$ where each node $i \in \mathbb{Z}_p$ uses a cyclically shifted distribution, shifting the assignment of sampling position $k$ to class $c$ by $(c + \lfloor i \cdot C/p + 0.5 \rfloor) \mod C$. We then partition the global dataset class-wise, assigning each node a fraction of the samples labeled with that class according to their relative frequency compared to other nodes. Depending on the global class imbalance, this may not result in the exact local class frequencies prescribed by $\mu_{\text{partition}}$, but rather the closest match to partition the global dataset according to these weights. It can also lead to an imbalance in the size of the local datasets across nodes, as requesting a large share from the majority class results in more samples than a large share of a minority class. Figure 8(b) illustrates the resulting global and local class frequencies for all four combinations of $\mu_{\text{data}} = \{\infty, 2\}$ and $\mu_{\text{partition}} = \{\infty, 2\}$.

We evaluate the predictive performance of distributed random forests trained on distributed data with different factors of imbalance. We train three dataset sizes **100K** with $t = 224$ trees, **1M** with $t = 800$, and **10M** with $t = 224$ trees on $p = 16$ compute nodes, partitioning and distributing both trees and training data over the nodes. Using the process described above, we generate global datasets ranging from no class imbalance ($\mu_{\text{data}} = \infty$) to a strong imbalance of up to $\mu_{\text{data}} = 0.5$. These global datasets are then partitioned and distributed among the 16 nodes, using varying partition imbalances $\mu_{\text{partition}}$, which affect the difference in data distribution across the 16 local subforests. Overall, we evaluate six values $\mu = \{\infty, 10, 5, 2, 1, 0.5\}$ for both $\mu_{\text{data}}$ and $\mu_{\text{partition}}$, resulting in a total of 36 combinations per dataset size. To avoid the over-representation of majority classes, we use balanced accuracy (macro average). Figure 9 gives the balanced accuracy on three datasets **100K**, **1M**, and **10M**, while varying both the data and partition imbalance from $\infty$ to 0.5. Additional metrics are given in Figures 10 and 11. The imbalance of the global dataset is shown on the x-axis, with the imbalance increasing as $\mu_{\text{data}}$ decreases. As expected, the classification gets increasingly harder, and the balanced accuracy decreases with a stronger imbalance. This applies across all three datasets and for all partition imbalances. Interestingly, increasing the partition imbalance (by decreasing $\mu_{\text{partition}}$) can actually improve the predictive performance of the global model. For imbalanced global datasets with $\mu_{\text{data}} \leq 10$, the accuracy consistently improves as the local imbalance increases from $\mu_{\text{partition}} = \infty$ to $\mu_{\text{partition}} = 1$. For a balanced global dataset $\mu_{\text{data}} = \infty$, we observe almost no difference between $\mu_{\text{partition}} = \infty$ to $\mu_{\text{partition}} = 1$. Only $\mu_{\text{partition}} = 0.5$ performs significantly worse than $\mu_{\text{partition}} = \infty$ for some cases, especially for datasets with fewer samples and more balanced global class frequency. This suggests that there is a limit to how imbalanced the local subsets may be before performance degrades. Overall, our results suggest that some class imbalance in the local datasets has only a small impact on accuracy when the global dataset is balanced and may even improve performance on imbalanced global data. Appendix A.5 investigates this effect in more detail by comparing local and global confusion matrices for IID and non-IID partitions. An important limitation of this experiment is the restriction to synthetic data; however, this allows us to vary the dataset size and imbalance freely, focusing purely on the effects of non-IID data without side effects. Repeating these experiments with real-world data will be an important next step in evaluating the practical implications of these observations.

## 6 Conclusion

This paper studies the scalability of distributed random forests. By distributing independent subforests across compute nodes and utilizing shared-memory parallelization within each node, we implement a hybrid parallel approach to distributed random forests. We examine the parallel scalability of this approach in both strong and weak scaling experiments, scaling up to 64 compute nodes with 76 cores each, to a total of 4864 cores. We achieve strong scaling speedups of up to 31.98, which level out as the work per node diminishes, and weak scaling efficiencies above 0.96 for all tested scales. The parallelization does not impact the random forest's predictive performance and the accuracy of the global model with $t$ trees remains unchanged when increasing the compute nodes $p$. In contrast, the mean accuracy of the local models decreases with $p$ as their size $t/p$ decreases. We compare two approaches to inference using either a distributed model with global voting or aggregating the global model on a single node, followed by sequential inference. The global voting approach benefits from low memory consumption per node and faster inference at the cost of using more

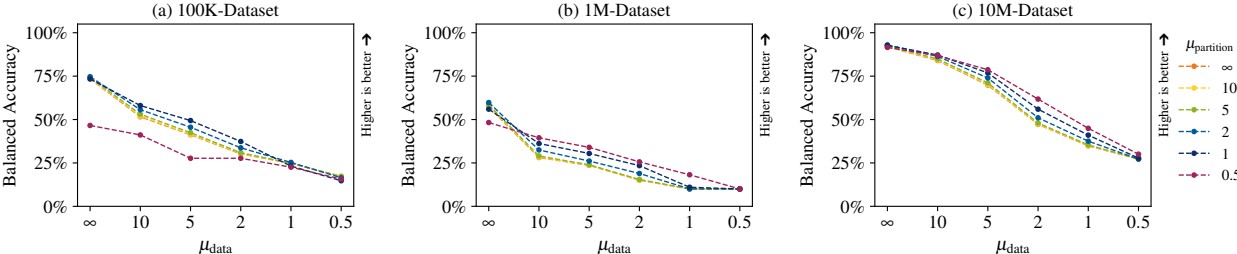

Figure 9: The balanced accuracy for different combinations of class imbalance in the global data ($\mu_{\text{data}}$) and the local subsets ($\mu_{\text{partition}}$) using three different problem sizes: **100K** with $t = 224$ trees, **1M** with $t = 800$, and **10M** with $t = 224$ trees on $p = 16$ compute nodes. The smaller $\mu$, the stronger the imbalance. Increasing the imbalance of the global data consistently decreases predictive performance, while an increased partition imbalance can improve performance on imbalanced global data.

compute nodes. In contrast, aggregating a global model introduces a computational and memory overhead, yet requires only a single node and no sharing of inference data. We further investigate how model size, the number of training samples, their partitioning, and class distribution impact the predictive performance of large-scale random forests. As expected, increasing both the number of trees and the number of training samples improves results. However, partitioning the training data, for example, across distributed machines, has a significant impact on the resulting accuracy. In testing various combinations of class imbalance in the global data and the partition into local subsets, we find that increasing global imbalance complicates the classification of minority classes. Introducing a partition imbalance can counteract this increasing diversity between subforests. This work improves the understanding of the performance characteristics of distributed random forests and their limitations at scale. While the training scales near perfectly, given enough work is available, inference involves a tradeoff between memory consumption, computation time, and data privacy. Our results on data distribution and heterogeneity provide key insights for applying distributed ensembles like random forests in practice and in the context of federated learning. In the future, we aim to extend our experiments to real-world datasets, non-classification tasks like regression, and a broader range of ensemble methods.

### Acknowledgments

This work was funded by Helmholtz Association's Initiative and Networking Fund through the Helmholtz AI platform grant. The authors gratefully acknowledge the computing time provided on the high-performance computer HoreKa by the National High-Performance Computing Center at KIT (NHR@KIT). This center is jointly supported by the Federal Ministry of Education and Research and the Ministry of Science, Research and the Arts of Baden-Württemberg, as part of the National High-Performance Computing (NHR) joint funding program (https://www.nhr-verein.de/en/our-partners). HoreKa is partly funded by the German Research Foundation (DFG).

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

# A  Appendix

## A.1  Distributed Random Forests – Pseudocode

Algorithm 1 summarizes the tree-parallel training of distributed random forests. Algorithms 2 and 3 describe the two variants for inference, either aggregating a global model or using global voting for distributed inference.

---

**Algorithm 1:** Training of local subforest on each compute node.

**Input:** Local training data $D_{\text{local}}$, number of global trees $t_{\text{global}}$, number of compute nodes $p$

**1** $t_{\text{local}} \leftarrow t_{\text{global}}/p$             ▷ Global forest distributed over nodes

**2** $F_{\text{local}} \leftarrow \text{randomforest}(t_{\text{local}})$       ▷ Create independent local subforest

**3** **foreach** $T \in F_{\text{local}}$ **do in parallel**       ▷ Shared-memory parallelization within each node

**4**      $\text{grow}(T, D_{\text{local}})$

---

**Algorithm 2:** Aggregation of the global model and local inference.

**Input:** Local inference data $D$, local forest $F_{\text{local}}$

**1** $F_{\text{global}} \leftarrow \texttt{all\_gather}(F_{\text{local}})$       ▷ Collect global model, only once after training

**2** **return** $F_{\text{global}}.\text{predict}(D)$       ▷ Local inference on global model

---

**Algorithm 3:** Inference with the distributed model via global voting.

**Input:** Shared inference data $D$, local forest $F_{\text{local}}$

**1** $h_{\text{local}} \leftarrow F_{\text{local}}.\text{predict\_histogram}(D)$       ▷ Collect local predictions as histogram

**2** $h_{\text{global}} \leftarrow \texttt{all\_reduce}(h_{\text{local}})$       ▷ Sum local histograms

**3** **return** $\text{argmax}(h_{\text{global}})$       ▷ Global prediction is the most-voted class

---

## A.2  Experimental Setup

### A.2.1  Implementation

We implement the distributed random forest in `Python` using `scikit-learn` and MPI. Each node-local subforest is a `scikit-learn RandomForestClassifier` and trained via the `fit` method, passing in local training data and labels to train each individual decision tree in the ensemble. For shared-memory parallelization, we enable `scikit-learn`'s parallelization in the local subforests via the `n_jobs` parameter. All-gathering the global model and aggregating the local histograms for global voting is implemented in `mpi4py`. We train random forests of different sizes, varying the number of trees $t$ but keeping to the default hyperparameters defined by `scikit-learn`. Our code is open-source and publicly available at github.com/Helmholtz-AI-Energy/special-couscous.

### A.2.2  Computational Environment

All experiments were conducted on up to 64 compute nodes, each of which has two Intel Xeon Platinum 8368 processors for a total of 76 cores, 64 kB L1 and 1 MB L2 cache per core, and 57 MB L3 cache per processor. Most experiments used standard compute nodes with 256 GB main memory. The exception is the serial baseline for the strong scaling experiments, which used high-memory nodes with 512 GB main memory but otherwise identical hardware to fit the model and data. All nodes are connected with InfiniBand 4X HDR 200 Gbit/s interconnect. All experiments used `OpenMPI` v4.1.6, `Python` v3.11.2, `mpi4py` v4.0.1, `numpy` v2.2.2, `scikit-learn` v1.6.1, and `scipy` v1.15.1.

## A.3 Detailed Results on Scaling Laws of Random Forests

Tables 2 and 3 give the results illustrated in Figure 1 in tabular format.

Table 2: The predictive performance in test accuracy when scaling the total number of trees $t = \lambda \cdot t_1$ for two baselines **1M** with $t_1 = 25$ and **10M** with $t_1 = 7$.

| | 1M | | | | 10M | | | |
|---|---|---|---|---|---|---|---|---|
| $\lambda$ | $n$ | $t_1$ | Trees $t$ | Accuracy ↑ | $n$ | $t_1$ | Trees $t$ | Accuracy ↑ |
| 1 | $7.5 \times 10^5$ | 25 | 25 | 0.136 | $7.5 \times 10^6$ | 7 | 7 | 0.490 |
| 2 | $7.5 \times 10^5$ | 25 | 50 | 0.152 | $7.5 \times 10^6$ | 7 | 14 | 0.632 |
| 4 | $7.5 \times 10^5$ | 25 | 100 | 0.176 | $7.5 \times 10^6$ | 7 | 28 | 0.767 |
| 8 | $7.5 \times 10^5$ | 25 | 200 | 0.213 | $7.5 \times 10^6$ | 7 | 56 | 0.862 |
| 16 | $7.5 \times 10^5$ | 25 | 400 | 0.271 | $7.5 \times 10^6$ | 7 | 112 | 0.915 |
| 32 | $7.5 \times 10^5$ | 25 | 800 | 0.357 | $7.5 \times 10^6$ | 7 | 224 | 0.941 |
| 32 | $7.5 \times 10^5$ | 25 | 800 | 0.358 | $7.5 \times 10^6$ | 7 | 224 | 0.941 |
| 64 | $7.5 \times 10^5$ | 25 | 1600 | 0.476 | $7.5 \times 10^6$ | 7 | 448 | 0.953 |
| 64 | $7.5 \times 10^5$ | 25 | 1600 | 0.476 | $7.5 \times 10^6$ | 7 | 448 | 0.953 |
| 128 | $7.5 \times 10^5$ | 25 | 3200 | 0.613 | $7.5 \times 10^6$ | 7 | 896 | 0.958 |
| 256 | $7.5 \times 10^5$ | 25 | 6400 | 0.736 | $7.5 \times 10^6$ | 7 | 1792 | 0.961 |
| 512 | $7.5 \times 10^5$ | 25 | 12 800 | 0.820 | $7.5 \times 10^6$ | 7 | 3584 | 0.962 |
| 1024 | $7.5 \times 10^5$ | 25 | 25 600 | 0.867 | $7.5 \times 10^6$ | 7 | 7168 | 0.963 |
| 2048 | $7.5 \times 10^5$ | 25 | 51 200 | 0.892 | $7.5 \times 10^6$ | 7 | 14 336 | 0.963 |

Table 3: The predictive performance in test accuracy when scaling the total number of trees $t = \lambda \cdot t_1$ with or without data scaling. With data scaling, the number of training samples grows with $\lambda$ as $n = \lambda \cdot n_1$. Without data scaling, the training set does not grow with $\lambda$, remaining at a constant $n = n_1$.

| | | | | **With Data Scaling** | | **Without Data Scaling** | |
|---|---|---|---|---|---|---|---|
| $n_1$ | $t_1$ | $\lambda$ | Trees $t$ | $n$ | Accuracy ↑ | $n = n_1$ | Accuracy ↑ |
| $1.2 \times 10^4$ | 25 | 1 | 25 | $1.2 \times 10^4$ | 0.111 | $1.2 \times 10^4$ | 0.111 |
| $1.2 \times 10^4$ | 25 | 2 | 50 | $2.3 \times 10^4$ | 0.116 | $1.2 \times 10^4$ | 0.113 |
| $1.2 \times 10^4$ | 25 | 4 | 100 | $4.7 \times 10^4$ | 0.129 | $1.2 \times 10^4$ | 0.117 |
| $1.2 \times 10^4$ | 25 | 8 | 200 | $9.4 \times 10^4$ | 0.165 | $1.2 \times 10^4$ | 0.127 |
| $1.2 \times 10^4$ | 25 | 16 | 400 | $1.9 \times 10^5$ | 0.214 | $1.2 \times 10^4$ | 0.145 |
| $1.2 \times 10^4$ | 25 | 32 | 800 | $3.8 \times 10^5$ | 0.315 | $1.2 \times 10^4$ | 0.160 |
| $1.2 \times 10^4$ | 25 | 64 | 1600 | $7.5 \times 10^5$ | 0.474 | $1.2 \times 10^4$ | 0.188 |
| $1.2 \times 10^5$ | 7 | 1 | 7 | $1.2 \times 10^5$ | 0.276 | $1.2 \times 10^5$ | 0.276 |
| $1.2 \times 10^5$ | 7 | 2 | 14 | $2.3 \times 10^5$ | 0.415 | $1.2 \times 10^5$ | 0.357 |
| $1.2 \times 10^5$ | 7 | 4 | 28 | $4.7 \times 10^5$ | 0.588 | $1.2 \times 10^5$ | 0.471 |
| $1.2 \times 10^5$ | 7 | 8 | 56 | $9.4 \times 10^5$ | 0.767 | $1.2 \times 10^5$ | 0.600 |
| $1.2 \times 10^5$ | 7 | 16 | 112 | $1.9 \times 10^6$ | 0.882 | $1.2 \times 10^5$ | 0.721 |
| $1.2 \times 10^5$ | 7 | 32 | 224 | $3.8 \times 10^6$ | 0.933 | $1.2 \times 10^5$ | 0.810 |
| $1.2 \times 10^5$ | 7 | 64 | 448 | $7.5 \times 10^6$ | 0.953 | $1.2 \times 10^5$ | 0.862 |

### A.4 Detailed Results on Parallel Scalability

Tables 4 and 5 give the results presented in Section 5 in tabular format.

Table 4: Comparison of the two inference variants either aggregating the global model or using distributed inference via global voting in a weak scaling setup, giving the parallel training time $T_{\text{train}}(p)$, time to aggregate the global model $T_{\text{gather}}(p)$, and time to perform inference on the entire test set $T_{\text{test}}(p)$. Note that the global model performs sequential inference on a single node, while the inference with global voting is parallelized across $p$ nodes. We further include the memory consumption (Mem) per node. When aggregating a global model, the memory consumption per node scales with the global model size $t$, while with global voting, it scales only with the local model size $t/p$.

| | $p$ | $t$ | $t/p$ | Global Model | | | | Global Voting | | |
|---|---|---|---|---|---|---|---|---|---|---|
| | | | | $T_{\text{train}}(p)$ ↓ | $T_{\text{gather}}(p)$ ↓ | $T_{\text{test}}(p)$ ↓ | Mem ↓ | $T_{\text{train}}(p)$ ↓ | $T_{\text{test}}(p)$ ↓ | Mem ↓ |
| 100K | 2 | 152 | 76 | 5.776 s | 0.550 s | 0.179 s | 3.3 GiB | 5.677 s | 0.110 s | 2.5 GiB |
| | 4 | 304 | 76 | 5.588 s | 3.673 s | 0.307 s | 4.2 GiB | 5.584 s | 0.115 s | 2.6 GiB |
| | 8 | 608 | 76 | 5.677 s | 6.052 s | 0.558 s | 6.0 GiB | 5.632 s | 0.123 s | 2.6 GiB |
| | 16 | 1216 | 76 | 5.753 s | 7.756 s | 1.059 s | 9.7 GiB | 5.739 s | 0.112 s | 2.5 GiB |
| | 32 | 2432 | 76 | 5.944 s | 14.592 s | 2.047 s | 17.0 GiB | 6.022 s | 0.123 s | 2.6 GiB |
| | 64 | 4864 | 76 | 6.412 s | 23.226 s | 4.015 s | 31.5 GiB | 6.514 s | 0.122 s | 2.8 GiB |
| 1M-b | 2 | 152 | 76 | 30.119 s | 4.748 s | 0.908 s | 10.4 GiB | 30.396 s | 0.523 s | 6.4 GiB |
| | 4 | 304 | 76 | 30.244 s | 6.135 s | 1.672 s | 14.9 GiB | 30.100 s | 0.527 s | 8.1 GiB |
| | 8 | 608 | 76 | 30.051 s | 13.004 s | 3.206 s | 24.2 GiB | 30.002 s | 0.545 s | 8.6 GiB |
| | 16 | 1216 | 76 | 30.327 s | 23.451 s | 6.313 s | 44.2 GiB | 30.280 s | 0.545 s | 6.7 GiB |
| | 32 | 2432 | 76 | 30.519 s | 42.331 s | 12.504 s | 82.6 GiB | 30.632 s | 0.546 s | 7.0 GiB |
| | 64 | 4864 | 76 | 31.089 s | 115.327 s | 25.889 s | 159.8 GiB | 31.016 s | 0.546 s | 6.6 GiB |

Table 5: The global training time $T(p)$, speedup $S(p)$, and accuracy for $p = 1, \ldots, 64$ nodes when distributing the training data, i.e., each node trains $t/p$ trees on $n/p$ local training samples.

| | Nodes $p$ | Trees $t$ | $t/p$ | $n/p$ | $T(p)$ ↓ | $S(p)$ ↑ | Accuracy ↑ |
|---|---|---|---|---|---|---|---|
| 1M | 1 | 1600 | 1600 | $7.5 \times 10^5$ | 5923.104 s | 1.000 | 0.476 |
| | 2 | 1600 | 800 | $3.8 \times 10^5$ | 1361.426 s | 4.351 | 0.420 |
| | 4 | 1600 | 400 | $1.9 \times 10^5$ | 331.704 s | 17.857 | 0.365 |
| | 8 | 1600 | 200 | $9.4 \times 10^4$ | 79.640 s | 74.373 | 0.315 |
| | 16 | 1600 | 100 | $4.7 \times 10^4$ | 22.113 s | 267.854 | 0.269 |
| | 32 | 1600 | 50 | $2.3 \times 10^4$ | 5.848 s | 1012.908 | 0.232 |
| | 64 | 1600 | 25 | $1.2 \times 10^4$ | 2.534 s | 2337.587 | 0.196 |
| 10M | 1 | 448 | 448 | $7.5 \times 10^6$ | 8067.080 s | 1.000 | 0.953 |
| | 2 | 448 | 224 | $3.8 \times 10^6$ | 1331.368 s | 6.059 | 0.942 |
| | 4 | 448 | 112 | $1.9 \times 10^6$ | 350.751 s | 22.999 | 0.929 |
| | 8 | 448 | 56 | $9.4 \times 10^5$ | 86.204 s | 93.581 | 0.913 |
| | 16 | 448 | 28 | $4.7 \times 10^5$ | 28.089 s | 287.201 | 0.892 |
| | 32 | 448 | 14 | $2.3 \times 10^5$ | 11.861 s | 680.150 | 0.868 |
| | 64 | 448 | 7 | $1.2 \times 10^5$ | 5.584 s | 1444.747 | 0.838 |

## A.5 Additional Details on the Effects of Class Imbalance

Figures 10 and 11 give the predictive performance with differing class imbalance using two additional metrics. The micro average of the accuracy weighs each sample with the same importance, thus placing a greater importance on majority classes. This effect can be observed in Figure 10 where the accuracy increases at strong imbalance as the majority class makes up an increasing share of all samples, especially in subfigure (b). Figure 11 gives the Cohen's kappa score

$$\kappa = \frac{p_o - p_e}{1 - p_e}$$

comparing the observed agreement $p_o$ between prediction and true label to the expected agreement $p_e$ based on the underlying class imbalance.

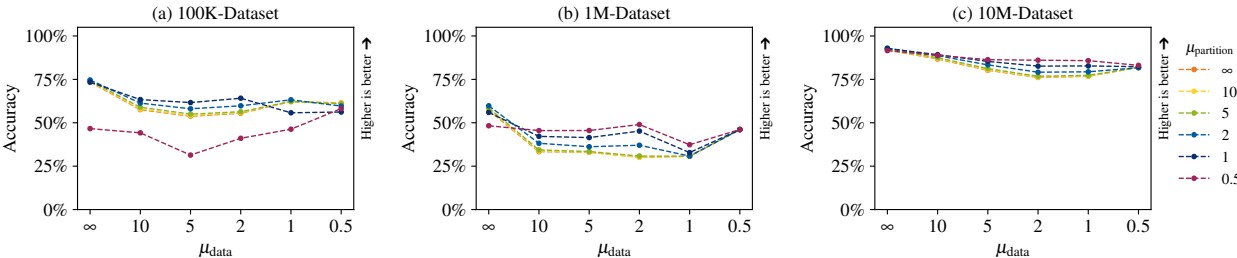

Figure 10: The (micro-average) accuracy for different combinations of class imbalance in the global data ($\mu_{\text{data}}$) and the local subsets ($\mu_{\text{partition}}$).

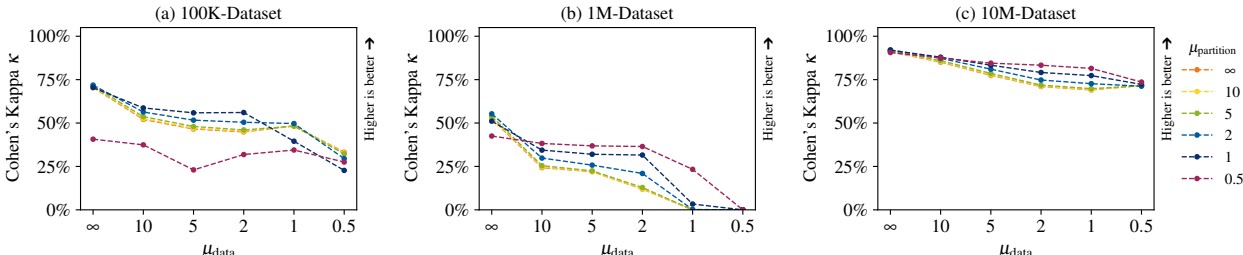

Figure 11: The Cohen's Kappa $\kappa$ for different combinations of class imbalance in the global data ($\mu_{\text{data}}$) and the local subsets ($\mu_{\text{partition}}$).

To further investigate the beneficial effect of partition imbalance, we provide a more detailed examination of the predictive performance on imbalanced data. Figures 12 and 13 give global and local confusion matrices for $\mu_{\text{data}} = 2$, comparing a balanced partition with $\mu_{\text{partition}} = \infty$ and an imbalanced partition with $\mu_{\text{partition}} = 1$. Figure 12 presents the confusion matrix of the global model and an exemplary local model (for node eight, using a balanced data partition). While the majority classes are retrieved mostly correctly, the global model struggles to detect the minority classes. As the data partition is balanced, all subforests are trained using the same class imbalance (that of the global data $\mu_{\text{data}}$). As a result, all local subforests fail to detect the rare classes. Figure 13 gives the corresponding global and local confusion matrices, but using an imbalanced data partition with $\mu_{\text{partition}} = 1$. The class distribution used to train the local subforests thus varies significantly from the global distribution. While this can reduce the predictive performance of individual subforests on the global test set, it strengthens the performance of the collective ensemble. In contrast to the balanced data partition, a non-IID data partition can increase the diversity between the local subforests, thereby improving the performance of the global ensemble. Table 6 gives the predictive performance of the global and local models with $\mu_{\text{data}} = 2$ and $\mu_{\text{partition}} = \infty$ compared to $\mu_{\text{partition}} = 1$.

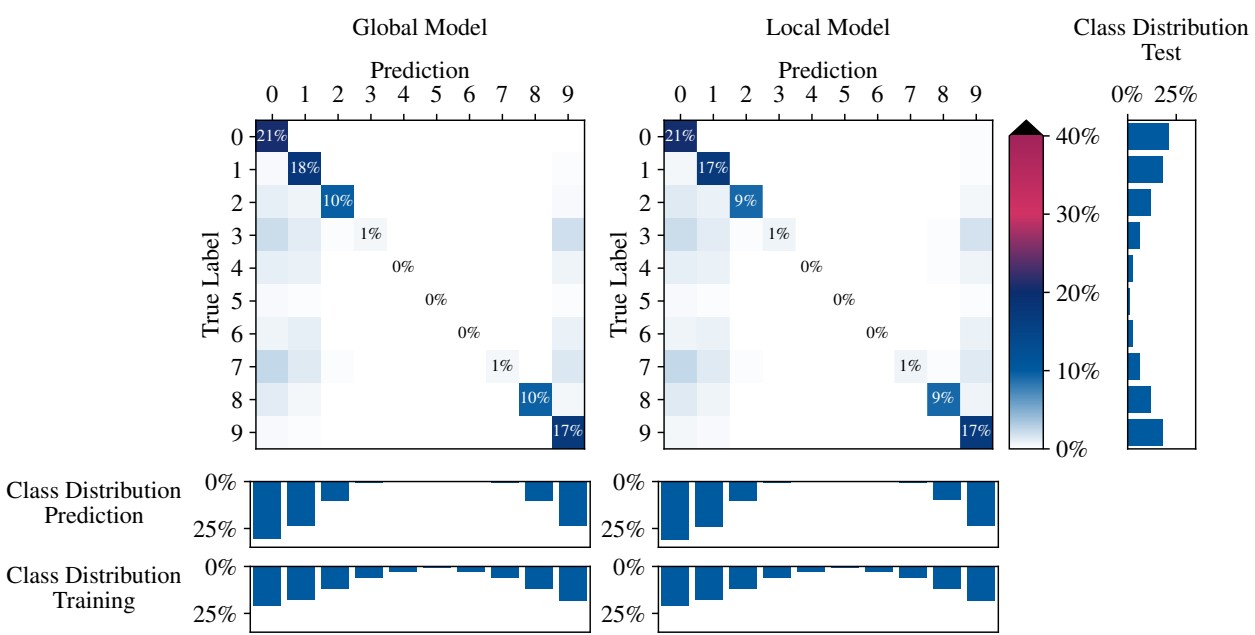

Figure 12: The confusion matrix of the global model (left) and the local model on node eight (right) using imbalanced data with $\mu_{\text{data}} = 2$ and balanced data partition with $\mu_{\text{partition}} = \infty$. We also provide the class distribution of the global test set, the global and local training sets, and the predictions of the global and local models. The class distribution of the local model is identical to the global distribution, and both fail at detecting the minority classes 3 to 7.

Table 6: The predictive performance with $\mu_{\text{data}} = 2$ and IID ($\mu_{\text{partition}} = \infty$) or non-IID ($\mu_{\text{partition}} = 1$) data for both the global model and as average over all local models. While the local models trained on non-IID data perform worse individually, their ensemble outperforms the training on IID data thanks to increased diversity between the subforests.

| | $\mu_{\text{partition}} = \infty$ | | $\mu_{\text{partition}} = 1$ | |
| Metric | Global Model | Local Models | Global Model | Local Models |
|---|---|---|---|---|
| Accuracy | 0.761 | 0.745 | 0.826 | 0.432 |
| Balanced Accuracy | 0.473 | 0.462 | 0.560 | 0.331 |
| Cohen's Kappa | 0.711 | 0.691 | 0.791 | 0.346 |

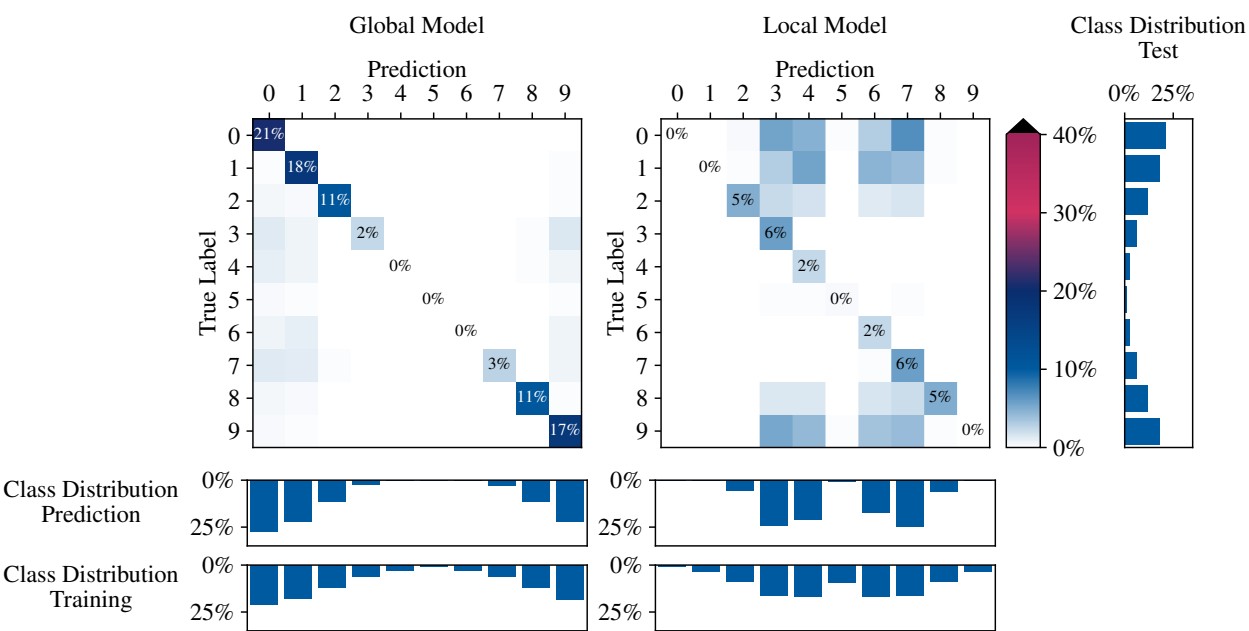

Figure 13: The confusion matrix of the global model (left) and the local model on node eight (right) using imbalanced data with $\mu_{\text{data}} = 2$ and an imbalanced data partition with $\mu_{\text{partition}} = 1$. We also provide the class distribution of the global test set, the global and local training sets, and the predictions of the global and local models. The class distribution of the local model differs significantly from the global distribution. The increased diversity between the local models slightly improves the detection of minority classes compared to the balanced partition in Figure 12.

