# OpenReview forum: "Scaling Laws of Distributed Random Forests"
_TMLR — Accepted by TMLR_

### Review · Reviewer_FoSJ · 2025-08-03

**Summary Of Contributions:**

This paper provides a thorough review, both in literature and via experiments, of parallel scaling for learning random forests on synthetic data.  There were not new algorithms presented, and the resulting experiments seem fairly standard.  Yet, this measured and fairly comprehensive report provides interesting results.


### Strengths:
 + The paper provides a really broad review of different attempts (many recent) in the literature of parallelizing the learning of random forests.  This part alone makes the paper useful for TMLR readers.

 + The experimental evaluation considers several ways of scaling, with different data sizes in data set size and dimension, and synthetic data class balance.

 + I found the results interesting (although perhaps for different reasons than the authors).

 + The writing was well done, very thorough and efficient as it explained many experiments.

### Weaknesses:
 - The writing and prose overly focuses on the scaling concerns, and not enough on the effect on accuracy.  If accuracy of an approach drops by even 10%, it is going to be very hard to convince someone to use it in a meaningful prediction situation-- even if they have to wait 10 or 100 times a long.  Strong parallel scaling effects that leads to a noticeable drop in accuracy (of say 5%) compared to a global model should be strongly discounted.
  (the paper is right that there may be privacy or distribution shift reasons that could explain this away -- but I'd say these are often secondary to the iid test accuracy.)

 - I would have liked to have seen discussion of XGBoost, and related methods that build trees dynamically -- and seemingly sequentially.  Random Forests have the most consistent benefits in this model.  However, because of the boosted nature, it appears much hard to parallelize.

 - The fourth line of the paper claims that random forests are "interpretable."   This is not a common view.  Maybe decision trees are (which are the base unit of random forests), but random forests then blend them together in a complicated way, so that the effect of individual components is lost in the mix.  They are an important learning model nonetheless, so I would just remove this justification.

 - The experiments are entirely from synthetic data.  Sometimes the method of generating synthetic data has quirks that is not reflected in real data.  It would be useful to run a few simple experiments to see if some of the main results are mirrored on some similarly scale real data sets -- in particular in how the accuracy is affected.



### Overall:
I liked the paper and found it interesting and useful to read.  It provides a nice review, and measured evaluation.  It leaves open some big issues that could improve the paper and its impact (see **Weaknesses**) but is already a self-contained paper.

**Audience:**

Yes

**Audience Explanation:**

I found reading the results interesting.  Generic random forests can scale fairly well via parallelism, but splitting data or a lack of strong global optimization can lead to drastic drops in accuracy.  This was demonstrated believably in the experimental results.

**Claims And Evidence:**

Yes

**Claims Explanation:**

The paper does not make strong claims, it presents a review.  I think the review is thorough and balanced.

**Requested Changes:**

The first and third weaknesses should definitely be addressed in the writing.

The second weakness (on XGBoost, and relatives) is something that could be shown as a baseline, and I would find it useful.  If it does not show much improvement over the synthetic data, it may indicate that the data inputs used are not very interesting.
I expect scaling XGBoost would require more advanced techniques than this paper explores to understand how it could be scaled effectively.

---

> ### Author Response · Authors · 2025-09-08
>
> **Regarding Accuracy**
> We would be grateful if you could indicate the specific subsection/figure in which you would like a more in-depth discussion of accuracy. Just to clarify: in the strong-scaling experiments (Section 5.2, Figure 3), the accuracy of the final/global model remains constant with increasing parallelization. We include the accuracy of the local models (which does decline as the local models reduce in size) because it highlights the differences between the strong and weak scaling setups, but the local models would not typically be used individually in practice.
> When distributing the data (Section 5.4, Figure 6), the accuracy does indeed decrease as the number of nodes increases. This is because the training data is partitioned across more nodes, and each tree thus trained on less data. This experiment is therefore more motivated by scenarios in which the data cannot be shared, and Figure 6(a) illustrates the impact of such scenarios. Would you like us to expand the discussion of accuracy in Section 5.4?
>
> **Regarding XGBoost**
> As you mentioned, the sequential nature of boosting means that such models are unlikely to benefit from parallelization over the trees as explored in this paper. Since each tree corrects the error of the previous trees, tree $t$ depends on trees $t-1$ making the training inherently sequential in the tree dimension. There are approaches to parallelize XGBoost, but they typically focus on parallelizing the computation within individual trees rather than across trees, and could be seen as orthogonal to the approach discussed in this paper.
>
> **Regarding Interpretablity**
> While random forests are not interpretable per se, we would argue that they are still more interpretable than deep neural networks. In any way, this is not directly related to the paper itself, and we have updated line 4 to refer to decision trees instead.
>
> **Regarding Synthetic Data**
> We agree on the limitations of synthetic data and did therefore repeat the strong and weak scaling experiments on the HIGGS dataset [1]. With $1.1\times10^7$ samples and $28$ features, it is similar in size to the 10M synthetic dataset. We use $640$ trees for strong scaling and scale up from $10$ to $640$ trees in weak scaling. We have added our results to the supplementary material, subdirectory `higgs/`. The scaling behavior for both predictive and parallel performance matches our results on synthetic data. If you find these useful, we are happy to run them for the remaining seeds and include them in the paper.
>
> [1] D. Whiteson. "HIGGS," UCI Machine Learning Repository, 2014. [Online]. Available: https://doi.org/10.24432/C5V312.

---

> > ### Comment · Reviewer_FoSJ · 2025-09-09
> >
> > Regarding accuracy, I think what I found interesting is that the global models held-up much better than the local models.  The sections describe this all accurately, but this insight was not highlighted in the intro or conclusion as much as I would have.  For instance, the intro lists a contribution:
> > > A parallel scalability study on up to 64 compute nodes, achieving up to 31.98 strong scaling speedup
> > and over 0.96 weak scaling efficiency without affecting predictive performance.
> >
> > Which is fine.  But what I found revealing, and not highlighted, was this only worked for your well-engineered global models, and this did not hold for the local ones.  I think this is worth highlighting.
> >
> >
> > Yes, I think it would be worth having a short section showing that the patterns on synthetic data were supported by the real higgs data.
> >
> > The interpretability change in the intro is good.  thanks.
> >
> > It's too bad not to be able to include XGBoost, but I accept that this can be out of scope, and leave for future work.

---

> > > ### Author Response · Authors · 2025-09-09
> > >
> > > **Regarding Accuracy**
> > > Thank you for this clarification, we now have a better understanding of your point.
> > > We propose refining the highlighted statement in the abstract and introduction to "[...] without affecting predictive performance *of the global model*." and adding a short discussion of the accuracy of global and local models to the conclusion (after the speedup and efficiency numbers in line 6 of the conclusion):
> > >
> > > > The parallelization does not impact the random forest's predictive performance and the accuracy of the global model with $t$ trees remains unchanged when increasing the compute nodes $p$. In contrast, the mean accuracy of the local models decreases with $p$ as their size $t/p$ decreases.
> > >
> > > We hope that these changes satisfactorily address your concern and would include them in the next revision or make further adjustments if needed.

---

> > > > ### Comment · Reviewer_FoSJ · 2025-09-09
> > > >
> > > > yes, great.  thanks

---

### Review · Reviewer_b4LG · 2025-08-12

**Summary Of Contributions:**

### Summary

Random forests are a popular class of machine learning algorithms that are particularly amenable to parallelization. Training random forests in parallel can result in large efficiency gains, and may be required in settings such as federated learning with sensitive datasets. The paper generates synthetic data to examine the performance of random forests (a) as the number of parallel compute resources increases and (b) as both the data size and compute resources increase. The paper considers cases in which the distributed nodes share a global dataset, and where each node has a local subset of the data. Finally, the authors examine the case where the local datasets at each node have class imbalance.

### Strengths and weaknesses

*Imbalanced data*. I think the analysis of imbalanced data settings is particularly interesting for practical uses of distributed random forests. For example when hospitals with different datasets use federated random forests, disparities in accuracy for under-resourced locations or between classes would be a major concern. One concern is that in the imbalanced setting, the authors mention that their technique for sampling from $\mu_{partition}$ results in different dataset sizes between nodes, and the nodes with higher weights on the majority class tend to have larger sizes. I'm concerned that in real cases, the split between nodes would be constant, and that this correlation between local dataset size and class prevalence could interact with the results.

*Related work.* The related work is thorough and detailed. However, it is often unclear how the prior findings listed such as Eng and Park (2024), Mitchell et al. (2011), Vazquez-Novoa et al. (2023), etc, compare to the contributions of this paper: it's difficult to tell what is the contribution of the work at hand. An additional specific concern is that Mitchell et al.'s results appear to show that distributed random forests do not scale well for real data, which questions the generalizability of these synthetic experiments.


*Synthetic data*. This paper uses exclusively synthetic data. This is a strength for the imbalanced data case, because the authors can control the parameter $\mu$ of the data to directly assess the effect of class imbalance. However, in the remaining experiments the use of synthetic data does not seem to be well-justified, because without having some parameter of the distribution being controlled, it's not clear to me why the authors could not use real data for this.

Since the paper only considers a single distribution (independent normal distributions with variance 1) for the scaling results, it's difficult to have intuition for how these results would extend to more realistic distributions. If the variables interact in more complex ways -- as in real data -- the local dataset distributed settings would likely scale differently (likely more poorly).

The plots consider runs for only 3 random seeds and do not report standard error bars, so it's difficult to tell how significant the differences are between the curves (especially in the imbalanced setting where the curves are all very close together)

Furthermore, the test accuracy on the 1M dataset is no better than random in the best case, indicating that this may not be a realistic approximation for a setting in which random forests would be used.


*Implementation*. It would be helpful if the authors could clarify the sense in which the implementation is a contribution. Is the code released as a usable package that would easily work on other compute clusters? Furthermore, is the parallelization implemented here challenging to implement?

**Audience:**

Yes

**Audience Explanation:**

Yes, researchers training distributed random forests in federated learning settings would be interested in this work.

**Claims And Evidence:**

Yes

**Claims Explanation:**

Yes, on the whole the claims of the paper are backed by evidence. However, the paper does not report standard deviations / error bars for the plots shown. Moreover, the generalizability of the paper is limited as the random forests are trained on synthetic data from a single distribution.

**Requested Changes:**

Critical adjustments:
1. Please include error bars in your plots
2. Please either consider additional synthetic data distributions, or supplement the synthetic results with real data, or else provide a strong justification for why the results for this distribution would extend to real datasets
3. Please provide justification for why it is reasonable to have different-sized local datasets in the imbalanced class setting, and why this does not contribute to the surprising result that imbalanced local data can be helpful
4. Please clarify the relationship to related work
5. Please explain why it is reasonable to consider settings where the best-case test accuracy is as good as random.

Minor adjustments:
6. It would be helpful to clarify the sense in which the implementation is a novel contribution

---

> ### Author Response · Authors · 2025-09-08
>
> 1. **Error Bars:** We have added Figures with error bars to the supplementary material, subdirectory `error_bars/`. We initially omitted them because the errors are small but are happy to update the paper accordingly if you feel this would improve clarity.
> 2. **Synthetic Data:** We agree on the limitations of synthetic data and did therefore repeat the strong and weak scaling experiments on the HIGGS dataset [1]. With $1.1\times10^7$ samples and $28$ features, it is similar in size to the 10M synthetic dataset. We use $640$ trees for strong scaling and scale up from $10$ to $640$ trees in weak scaling. Our results are available in subdirectory `higgs/` in the supplementary. The scaling behavior for both predictive and parallel performance matches our results on synthetic data. If you find these useful, we are happy to run them for the remaining seeds and include them in the paper.
> 3. **Local Dataset Size in Non-IID Experiments:** We would argue that in real-world scenarios, both equal-sized and differently sized local datasets are plausible. In the hospital example, the number of patients—thus the number of samples collected—may vary substantially depending on each hospital's size, location, and specialization. It therefore seems unlikely for all hospitals to collect datasets of exactly the same size.
>    Nonetheless, it is indeed interesting to consider the setup you suggested where all local subsets contain the same number of samples. We therefore conducted an additional experiment, relaxing the constraints on the local class distributions while enforcing both the global class distribution and a constant subset size, using iterative proportional fitting (more details on the construction below). We repeat the non-IID experiment with this setup on the 100K dataset (Figure 8(a)); our results are available here: in subdirectory `non_iid__const_subset_size/` in the supplementary. In this setup, $\mu_\text{partition}$ has less impact on the results, and increased partition imbalance tends to slightly *decrease* performance. Notably, the results for $\mu_\text{partition}=\infty$ remain unchanged; that is, the more imbalanced partitions perform overall worse in this setup.
>    We are open to extend this experiment to the other datasets and incorporate this setup into Section 5.5.

---

> > ### Author Response · Authors · 2025-09-08
> >
> > 4. **Relationship to Related Work:**
> > 	- Antonio Eng Lim and Hee Park (2024): In our case, the trees are distributed over the nodes and trained individually (i.e., only one node contributes to each tree; the collaboration happens afterward by combining the local subforests to a global forest), whereas they grow each tree collaboratively (i.e., each node contributes to each tree). Due to collaborating on each tree, their approach has higher computational complexity and communication overhead. Their evaluation is focused on predictive performance. and they do not report runtime results or scalability over different numbers of nodes.
> > 	  In terms of non-IID data, they examine the effect of alpha chunking with $\alpha=2,\dots,6$ over ten nodes. Different $\alpha$ affect the local class balance, but they don't consider different global class balances or the interaction between local and global imbalance and are limited to the relatively small StatLog dataset with only 6.5k samples.
> > 	- Both Mitchell et al. (2011) and Vazquez-Novoa et al. (2023) are closely related to our experiments in Section 5.2, that is, examining parallel scalability when parallelizing over trees via strong and weak scaling. They focus purely on parallel performance and do not consider different inference approaches, distributed or non-IID data (i.e., our Sections 5.3 to 5.5), or how the accuracy scales of different numbers of nodes and trees.
> > 	  More specifically, Mitchell et al. (2011) focus on a very particular type of data: namely, microarray data containing gene expression of biological samples, where the number of features vastly outweighs the number of samples. In contrast, we consider the more common case where there are more samples than features. Could you clarify which of Mitchell et al. (2011)'s figures you are referring to with "Mitchell et al.'s results appear to show that distributed random forests do not scale well for real data"? Since they do not consider predictive performance and report only runtime and related measures, we presume you are referring to the diminishing returns of parallelization at very high scale (in their case, number of cores > 128 in their Figure 2). This strong scaling limit can indeed also be observed in our results (e.g., our Figure 3(c)), with the speedup saturating for large numbers of nodes (in our case, each node corresponds to 76 cores) as is to be expected from Amdahl's law (see e.g. [2]). The remaining work per node needs to be large enough to compensate for the parallelization overhead. We have done some additional experiments on the HIGGS dataset (file `strong_scaling_for_problem_size.pdf` in the supplementary) to demonstrate how the parallel scalability depends on the available work (in terms of number of trees) on a real-world example. We expect results on the synthetic datasets to be very similar and are happy to extend the plot to include them if required.
> > 	  Vazquez-Novoa et al. use a task-based master-worker paradigm with a much more fine-grained parallelization scheme—creating multiple tasks per node for each node in all trees. They focus on different task scheduling approaches (with and without nesting) and how to respond to potential task failures. Their approach is based on the COMPS programming model, which is not as widely used and available as MPI. Their evaluation uses up to 48 trees and 16 nodes, with speedups of up to 6, and is thus of significantly smaller scale than this paper, which scales up to 51k trees, 64 nodes, and speedups of up to 32.
> > 5. **1M in Non-IID Scenario:** We included the 1M dataset in Figure 8 for consistency with the remaining experiments since it only collapses to near-random performance at high imbalance factors. However, we are open to removing it and instead showing only 100K and 10M datasets in Figure 8, or alternatively adding a line indicating random performance for clarity.
> > 6. **Implementation:** The code will be released as an open-source repository. It relies on MPI and Python, which are available on most compute clusters, and should thus be easily transferable to other systems. We are also open to releasing the code as PyPI package if you consider that to be useful.
> >
> > [1] D. Whiteson. "HIGGS," UCI Machine Learning Repository, 2014. [Online]. Available: https://doi.org/10.24432/C5V312.
> > [2] Gustafson, John L. "Reevaluating Amdahl's law." Communications of the ACM 31.5 (1988): 532-533.

---

> > > ### Author Response · Authors · 2025-09-08
> > >
> > > **More details on the construction of equal-sized local datasets**
> > > When using identical but shifted local class distributions with same-sized local subsets, the global dataset would follow a (near) uniform class distribution.
> > > To combine different, parameterized global and local imbalances with constant local subset sizes, we apply iterative proportional fitting (IPF) to relax the local class distributions.
> > > Specifically, we generate the global class distribution $p_\text{global}:Z_C\to[0, 1]$ and $i=1,\dots,p$ local class distributions $p_i:Z_C\to[0, 1]$ as before using a Skellam distribution with $\sum_{c=1}^Cp(c)=1$ (with $Z$ standing for $\mathbb{Z}$ but which Markdown seems to be unable to render in the above).
> > > We use the local class distributions as input matrix $M$ to the IPF algorithm with entries $M_{ic}=p_i(c)$.
> > > We enforce the global class distribution by using $p_\text{global}(c)\cdot p$ as column sum for columns $c=1\dots,C$ and $1$ as row sum for all rows.
> > > Given these inputs, we use IPF (via the `ipfn` python package) to obtain an output matrix close to $M$ (the original local class distributions) while fulfilling the row and column sum constraints (constant subset size and global class distribution).

---

### Review · Reviewer_kMux · 2025-08-18

**Summary Of Contributions:**

The contributions are empirical.

1. Experimental study of scaling laws of random forests with varying model size and data size.
2.  An implementation of distributed random forests for HPC. This is validated and shows a speed-up.
3. An evaluation of the effects of data distribution and class imbalance.

**Additional Comments:**

I believe there are citations missing like:

1. "An Agnostic Approach to Federated Learning with Class Imbalance" Shen et al.

**Audience:**

No

**Audience Explanation:**

There are no findings or conclusions. This paper seems more like a technical report than a paper. The authors implemented a distributed RF implementation, and then run it on a couple of setups. I fail to see the utility of this for the ML community.

In all, no rate is given, and no law presented.
It is an empirical study, but, not even a fitting of the law w.r.t to the number of nodes is given.

**Broader Impact Concerns:**

No comment on this regard.

**Claims And Evidence:**

No

**Claims Explanation:**

I believe that to claim scaling laws, a significnatly more extensive experimental setup should be considered. In looking at the figures, the reader cannot see a trend or a power law.
Also, the authors do not present any sort of approximation or fitting to the power laws that predict how random forrest will work.

**Requested Changes:**

1. Remove paragraph 3.2. I believe it cuts the flow of the paper. The content of this can be moved to the caption of Figure 1.
2. Before Section 5, the authors explain: "For shared-memory parallelization [...] " That whole parragraph should be in the appendix.

---

> ### Author Response · Authors · 2025-09-08
>
> - We understand the concern that claiming scaling laws might require additional experiments and the fitting of power laws, and agree that the current title may not perfectly reflect the scope of the paper. To better align the title with the presented results, we would be open to renaming the paper, e.g., to "Scaling Behaviors of Distributed Random Forests," if you feel this would more accurately represent the paper.
> - As suggested, we have merged Section 3.2 into the caption of Figure 1 and moved the implementation details from Section 4 to the appendix.
> - We thank the reviewer for pointing us to the work by Shen et al.; we have now incorporated the citation into the paper.

---

### Decision · Action_Editor_eTZz · 2025-10-21

**Recommendation:** Accept as is

**Audience:**

Yes

**Audience Explanation:**

I believe this work to be of interest to the machine learning community.

**Claims And Evidence:**

Yes

**Claims Explanation:**

Contributions are empirical and thoroughly discussed.